# Intrinsic Interferon Signaling Regulates the Cell Death and Mesenchymal Phenotype of Glioblastoma Stem Cells

**DOI:** 10.3390/cancers13215284

**Published:** 2021-10-21

**Authors:** Sabbir Khan, Rajasekaran Mahalingam, Shayak Sen, Emmanuel Martinez-Ledesma, Arshad Khan, Kaitlin Gandy, Frederick F. Lang, Erik P. Sulman, Kristin D. Alfaro-Munoz, Nazanin K. Majd, Veerakumar Balasubramaniyan, John F. de Groot

**Affiliations:** 1Department of Neuro-Oncology, The University of Texas MD Anderson Cancer Center, 1515 Holcombe Blvd, Houston, TX 77030, USA; skhan15@mdanderson.org (S.K.); SSen3@mdanderson.org (S.S.); juanemmanuel@tec.mx (E.M.-L.); ksgandy@mdanderson.org (K.G.); KDAlfaro@mdanderson.org (K.D.A.-M.); NKMajd@mdanderson.org (N.K.M.); 2Department of Symptom Research, MD Anderson Cancer Center, The University of Texas, Houston, TX 770030, USA; RMahalingam@mdanderson.org; 3Tecnologico de Monterrey, Escuela de Medicina y Ciencias de la Salud, Ave. Morones Prieto 3000, Monterrey 64710, Mexico; 4Department of Pathology and Genomic Medicine, Houston Methodist Research Institute, Houston, TX 77030, USA; akhan5@houstonmethodist.org; 5Department of Neurosurgery, The University of Texas MD Anderson Cancer Center, 1515 Holcombe Blvd, Houston, TX 77030, USA; flang@mdanderson.org; 6Department of Radiation Oncology, New York University, New York, NY 10016, USA; erik.sulman@nyulangone.org; 7Department of Neuro-Oncology, University of California, San Francisco, CA 94143, USA

**Keywords:** glioblastoma, interferon signaling, apoptosis, glioma stem-like cell, STAT1, cell proliferation

## Abstract

**Simple Summary:**

Interferon signaling is mostly studied in the context of immune cells. However, its role in glioma cancer cells is unclear. This study aimed to investigate the role of cancer-cell-intrinsic IFN signaling in tumorigenesis in glioblastoma (GBM). We found that GSCs and GBM tumors exhibited differential cell-intrinsic type I and type II IFN signaling, and the high IFN/STAT1 signaling was associated with mesenchymal phenotype and poor survival in glioma patients. IFN-β exposure induced cell death in GSCs with intrinsically high IFN/STAT1 signaling, and this effect was abolished by inhibition of IFN/STAT1 signaling. A subset of GBM patients with high IFN/STAT1 may benefit from the IFN-β therapy.

**Abstract:**

Interferon (IFN) signaling contributes to stemness, cell proliferation, cell death, and cytokine signaling in cancer and immune cells; however, the role of IFN signaling in glioblastoma (GBM) and GBM stem-like cells (GSCs) is unclear. Here, we investigated the role of cancer-cell-intrinsic IFN signaling in tumorigenesis in GBM. We report here that GSCs and GBM tumors exhibited differential cell-intrinsic type I and type II IFN signaling, and high IFN/STAT1 signaling was associated with mesenchymal phenotype and poor survival outcomes. In addition, chronic inhibition of IFN/STAT1 signaling decreased cell proliferation and mesenchymal signatures in GSCs with intrinsically high IFN/STAT1 signaling. IFN-β exposure induced apoptosis in GSCs with intrinsically high IFN/STAT1 signaling, and this effect was abolished by the pharmacological inhibitor ruxolitinib and STAT1 knockdown. We provide evidence for targeting IFN signaling in a specific sub-group of GBM patients. IFN-β may be a promising candidate for adjuvant GBM therapy.

## 1. Introduction

Glioblastoma (GBM) is the most common, aggressive, and lethal primary brain tumor in adults. With standard care, the median survival of GBM is 12–15 months, and the disease kills more than 10,000 people annually in the United States alone [1]. GBM tumors are highly resistant to radiation therapy and chemotherapy; therefore, recurrence is inevitable with standard-of-care treatment [2]. Over the past decade, high-throughput genomic studies have revealed an intratumor heterogeneity of GBM at both the genetic and epigenetic levels, and have classified GBM tumors into distinct molecular subtypes, including classical, proneural, and mesenchymal [3]. These molecular subtypes have been associated with key genetic features that drive specific gene expression patterns [3]. Despite our improved understanding of the tumor heterogeneity and molecular pathology of GBM at the transcriptomic level, advances in the treatment of GBM have been limited. Tumor heterogeneity, the presence of rare treatment-resistant GBM stem-like cells (GSCs), and an immunosuppressive tumor microenvironment (TME) are among the features postulated to critically contribute to GBM treatment failure [4]. The molecular heterogeneity of GBM is recapitulated in GSCs, which are known for their clonal heterogeneity and plasticity [4,5]. These challenging features are likely to influence disease progression and response to various treatment modalities, including immunotherapies [6]. 

Although immunotherapies have been successful in the treatment of many solid tumors [7], results of several randomized clinical trials of immunotherapies in GBM such as targeted vaccines, oncolytic viruses, and checkpoint blockers have been disappointing [8]. The lack of treatment response to immunotherapy in GBM is likely due to factors such as a profound immunosuppressive environment, marked genetic and antigenic heterogeneity, and a paucity of GBM-infiltrating T cells (i.e., cold tumors) [9]. Thus, many challenges need to be overcome for successful clinical development of immunotherapies in GBM. In many cancers, including brain tumors, the efficacy of chemotherapy, radiation therapy, targeted immunotherapies, and oncolytic viruses depends on interferon (IFN) signaling for direct tumor-cell killing and/or to elicit indirect antitumor immune responses [10,11]. Thus, dysregulation of IFN signaling in tumor cells or in immune cells might be involved in response or resistance to the various treatments. 

IFNs are a group of signaling molecules belonging to the cytokine class of proteins, made and released by host cells in response to viral infection to intensify the defenses of the immune system in nearby cells [12]. There are three known types of IFN signaling, types I, II, and III, which are activated by their specific ligands [12]. Type I and type II IFNs activate both shared and distinct signal transducer and activator of transcription (STAT) complexes, which regulate a downstream signaling cascade [13]. The IFN/STAT1 pathway modulates the expression of classical IFN-regulated genes that have key immune effector functions and play crucial roles in the efficacy of cancer immunotherapies [11]. IFN/STAT1 signaling also modulates diverse cellular processes, such as proliferation, differentiation, and cell death, and plays a central role in innate and adaptive immunity [14,15].

IFN signaling plays an important role in central nervous-system-specific functions, such as removal of myelin debris, maintaining integrity of the blood–brain barrier, cell differentiation, and tumor development [16,17]. Autocrine activation of IFN signaling promotes immune escape in GBM [18] and is associated with poor survival in a subtype-specific manner [19]. The constitutive IFN signaling in GBM tumors influences the efficacy of oncolytic virus therapy [20]. IFN/STAT1 signaling also contributes to GBM cell stemness [21,22] and invasiveness via modulation of IFN-regulatory factors (IRFs) [23]. Furthermore, different pluripotent and multipotent stem cells express cell-type-specific groups of IFN-stimulated genes (ISGs) to mediate antiviral resistance and the evolution of pathogen resistance [24]. Similarly, IFN-mediated signaling modulates the stemness properties and treatment resistance of many other cancers, such as breast cancer [22,25], ovarian cancer [26], melanoma [27], and lung cancer [28]. The role of IFN/STAT1 signaling in cell death and the development of treatment resistance in GBM is important to decipher. In this study, we investigated how cancer-cell-intrinsic IFN/STAT1 signaling regulates cell proliferation, mesenchymal phenotypes, and apoptosis in GSCs. 

## 2. Material and Methods

### 2.1. The Cancer Genome Atlas (TCGA) In-Silico Analyses 

We extracted type I and II IFN genes from MsigDB [29] signatures HALLMARK INTERFERON ALPHA RESPONSE (97 genes) and HALLMARK INTERFERON GAMMA RESPONSE (200 genes). We used TCGA GBM RNA-seq and subtype data accessed through the Broad Institute GDAC firehose (https://gdac.broadinstitute.org/) (accessed on 22 February 2021) and the cBioPortal for Cancer Genomics (http://cbioportal.org) (accessed on 22 February 2021) [30,31]. We obtained a score for each IFN signature and sample by calculating the average expression of the genes belonging to each signature. For glioma subtypes, patients’ data were divided into 2 groups, low (Z score < −2), and high (Z score > 2) score. We performed a Kaplan–Meier survival analysis according to the low and high scores of the IFN signature groups. Similarly, the RNA-seq dataset of a cohort of GSCs was used to analyze the expression profile for type I and type II IFN signaling genes among GSCs. In addition, the GlioVis platform was used for analyzing *STAT1* expression by glioma grade and tumor and nontumor compartment using TCGA data, Gill data, and Ivy GAP data [32]. A correlation analysis between three mesenchymal markers (*CHI3L1* or *YKL40*, *CD44*, and *SERPINE1*) and three classical markers of IFN signaling (*STAT1*, *MX1*, and *ISG15*) was also performed using the GlioVis platform [32]. RNA-seq data were log2 transformed and quantile-normalized. All the statistical analyses were performed using R, unless otherwise specified.

### 2.2. GSC Cell Culture 

MDA-GSCs were isolated from patient-derived surgical specimens at the MD Anderson Cancer Center, and GSCs were grown as previously described [33] (we refer to MDA-GSCs as GSCs throughout the entire manuscript). The GSCs were maintained in suspension culture in Dulbecco’s Modified Eagle Medium (DMEM) supplemented with epidermal growth factor, basic fibroblast growth factor, 2% B-27, and antibiotics at 37 °C in a 5% CO_2_ atmosphere as described [34]. Cells were subcultured using Accutase (Sigma-Aldrich, St. Louis, MO, USA) to dissociate GSC spheres into single cells, and the cell culture medium was changed twice per week. Cell culture was routinely tested for mycoplasma contamination using the MycoAlert mycoplasma detection kit (Lonza, Basel, Switzerland). All GSC generation was approved by the institutional review board (Protocol # LAB04-0001) of The University of Texas MD Anderson Cancer Center.

### 2.3. Conditioned Medium (CM) Collection and Estimation of Cytokines and IFN Secretion

GSCs (0.5 × 10^6^ cells/mL) were cultured in DMEM/F-12 medium supplemented with epidermal growth factor, basic fibroblast growth factor, 2% B-27, and antibiotics in 12-well plates for 72 h. The CM was collected using centrifugation at 300× *g* for 5 min, followed by filtering through a 0.5 µM syringe filter, and stored at −80 °C until use. The levels of secreted cytokines and IFNs in the CM were measured using commercially available enzyme-linked immunosorbent assay (ELISA) kits per the manufacturer’s instructions.

### 2.4. Drug Treatments and Sample Collections 

The GSC culture was centrifuged in individual 15 mL tubes, and the cell pellet was suspended with 0.5 mL of Accutase and kept at 37 °C for 3–5 min. The GSC spheres or clusters were gently mixed using a pipette to form a single-cell suspension, and then dead cells were removed with cell-culture-grade phosphate-buffered saline (Corning) using a centrifuge (300× *g* for 5 min). The cell pellet was suspended in 1.0 mL of the complete media, and cells were counted using an automated cell counter (Beckman Coulter, Carlsbad, CA, USA). Indicated single GSCs were plated in 6-well plates (2–3 × 10^5^/well) using 2.0 mL of complete media and treated with the indicated concentrations of IFN-γ, IFN-β, or ruxolitinib for the indicated times. After the specified times, the cells were collected, washed with phosphate-buffered saline, and processed according to assay protocols.

### 2.5. Immunoblot Analysis 

At the end of the indicated experiments, cells were collected and washed with phosphate-buffered saline, lysed in an ice-cold lysis buffer, radioimmunoprecipitation assay buffer containing protease and phosphatase inhibitors for 30 min on ice, and intermittently vortexed 3 times. Protein samples were centrifuged at 16,000× *g* for 15 min at 4 °C, and supernatants were collected in 1.5 mL microcentrifuge tubes. The protein concentration was determined by bicinchoninic acid assay; 10–25 µg of proteins were separated by 4–12% gradient bis-tris plus gels (Thermo Fisher Scientific, Waltham, MA, USA) using 3-(N-morpholino) propanesulfonic acid (MOPS) running buffer in the gel electrophoresis. The separated protein was transferred on PVDF membrane, blocked with 5% non-fat dry milk and 2% BSA, and immunoblotting was performed for the proteins of interest.

### 2.6. RNA Isolation and Quantitative Real-Time Polymerase Chain Reaction (qPCR)

Total RNA from GSCs was isolated using the RNeasy kit (Qiagen, Hilden, Germany) and quantified using NanoDrop (Thermo Fisher Scientific). An equal amount of RNA was reverse-transcribed into cDNA using a high-capacity cDNA reverse-transcription kit (Applied Biosystems, Waltham, MA, USA). Gene expression levels were measured using an Applied Biosystem 7500 Fast Real-Time PCR System (Thermo Fisher Scientific) with SYBR Green (Sigma-Aldrich) or TaqMan (Thermo Fisher Scientific) master mix using specific primers for target genes, and the expression profile was calculated using the 2^-ddct^ method. Glyceraldehyde 3-phosphate dehydrogenase (*GAPDH*) was used as an internal gene control for the relative quantification of genes.

### 2.7. Apoptosis Analyses by Annexin-V Staining

The GSCs (0.5 × 10^6^ cells) were treated with IFN-β (1000 IU/mL) in 6-well plates for 48 h. A flow cytometric analysis for evaluating the apoptosis was performed by annexin-V and DAPI staining. For the apoptotic cell death analysis, cells were washed twice with phosphate-buffered saline and resuspended in 100 μL of annexin-V binding buffer, and stained with 2 μL of annexin-V-PE (BD Biosciences, San Jose, CA, USA) antibody for 15 min at ambient temperature. Then, cells were washed with binding buffer to remove unbound antibody and resuspended in 500 μL of annexin-V binding buffer containing 0.5 μg/mL DAPI. The apoptotic cells were analyzed at the MD Anderson Flow Cytometry and Cellular Imaging Facility, and at least 10,000 cells were analyzed using a Gallios flow cytometer (Beckman Coulter). The cells positive for annexin-V staining were considered early apoptotic cells (EAC), while cells positive for both the annexin-V and DAPI were considered late apoptotic cells (LAC), and the sum of both the EAC and LAC was considered the total apoptotic cells.

### 2.8. Evaluation of Cell Viability and Cell Growth

Cell viability or proliferation with ruxolitinib and IFN-β treatments was evaluated using a luminescence cell viability assay (CellTiter-Glo, Promega, Fitchburg, WI, USA) according to the manufacturer’s protocol. The GSCs (5 × 10^3^ per well) were cultured in a white 96-well plate and treated with the specific inhibitors for the indicated times. The relative cell viability was calculated using control (untreated) cells for each plate.

### 2.9. Single-Cell RNA-Seq Analysis

The gene expression matrix for analysis was obtained from Darmanis et al. (GSE84465) [35], Neftel et al. (GSE131928) [36], and Yu et al. (GSE117891) [37]. The classification of tumor and nontumor was adopted from the study of Caruso et al. [38]. The analysis was performed using Seurat v3.0 [39]. The normalized data function was used for the normalization with the default parameter. The variation across the cells was regressed out using the ScaleData function with default settings. For clustering analysis, 2000 highly variable genes were selected using the FindVariableGenes function, and the expression matrix was centered and scaled. Next, a principal component analysis was applied to generate 100 principal components, and the JackStraw function was used to select the significant principal components to be used for further clustering and dimensionality reduction. To identify clusters of transcriptionally similar cells, unsupervised clustering was employed using the FindClusters function with K-parameter set to 10 and the resolution set to 0.5. For dimensionality reduction, we used a Uniform Manifold Approximation and Projection (UMAP) method employed in Seurat.

### 2.10. STAT1 Knockdown in GSCs by CRISPR/Cas9 Technology 

GSCs (0.3 × 10^6^) were plated in a 6-well plate in the GSC growth media without antibiotics. Human STAT1 CRISPR/Cas9 knockdown (KD) plasmid (#sc-400086, 2.5 g) or control CRISPR/Cas9 (#sc-418922, 2.5 g) plasmid, and STAT1-HDR plasmid (#sc-400086-HDR, 2.5 g), all from Santacruz Biotechnology, were cotransfected with lipofectamine LTX plus reagent (Thermo Fisher) as per the manufacturer’s instructions for overnight. Transfection media was replaced with fresh media, and cells were allowed to grow for 3 days. STAT1 KD and vector control positive clones were selected with puromycin (1 µg/mL) for 7–10 days, followed by STAT1 KD confirmation by qPCR and immunoblotting.

### 2.11. Statistical Analyses

Data were representative of at least two independent experiments. GraphPad Prism software was used for graphs and statistical analyses. Comparisons between two groups were performed using a Student’s *t*-test; comparisons between more than two groups were analyzed by one-way analysis of variance with corresponding Tukey’s multiple comparison tests. If not indicated otherwise, analyses of significance were performed using two-tailed tests, and *p* < 0.05 was considered statistically significant.

## 3. Results 

### 3.1. GCSs and GBM Tumors Exhibited Differential Basal IFN Signaling

To test whether intrinsic IFN signaling displayed differential expression in GBM tumors, we constructed a metagene score of the most abundantly expressed type I and type II IFN signaling hallmark genes [29]. By using the metagene scores for type I and type II IFN-responsive genes, we queried IFN signaling in the TCGA RNA-seq datasets, and identified subsets of tumors with basal ‘low’ and ‘high’ IFN signatures, and about 50% of the tumors displayed high expression of IFN signature genes for both type I and type II signaling (Figure 1A,B). Furthermore, ~35% of type I IFN-high tumors and ~60% of type II IFN-high tumors were the mesenchymal subtypes (Figure 1A,B). The Ivy GBM Atlas dataset also showed upregulated IFN signature genes for the type I and type II IFN signaling in the perinecrotic zone, and hyperplastic blood vessels in the cellular tumor as compared to the leading edge (Appendix A). Interestingly, survival analysis revealed that high IFN signatures for both type I and type II IFN signaling were negatively associated with survival of GBM patients (Figure 1C). 

Next, we examined type I and type II intrinsic IFN signaling in a cohort of GSCs using RNA-seq datasets (Sulman et al., unpublished data). About 60–75% of the GSCs displayed high expression of IFN signature genes for both type I and type II signaling. Interestingly, all the type I and type II high-IFN signaling GSCs were associated with the mesenchymal glioma subtype (Figure 1D,E). In contrast, GSCs with low IFN signaling were associated with proneural or classical subtype (Figure 1D,E). We further validated our in silico analysis of IFN signaling using a Western blot (WB) by evaluating the expression of two classical markers (STAT1 and MX1) of IFN signaling in a panel of six GSCs. Our WB data showed high expression of phosphorylated STAT1 (p-STAT1, Y701), total STAT1 (t-STAT1), and MX1 in intrinsically active IFN-signaling GSCs (Figure 2A). Furthermore, we validated the IFN/STAT1 signaling in the same set of GSCs by evaluating the expression of several IFN signaling markers, such as *STAT1*, *MX1*, *IRF9*, *OAS2*, *IRF1*, and *IFI16* by qPCR (Figure 2B–G). Overall, these results suggested that a subset of GBM tumors and GSCs exhibited constitutive activation of type I and type II IFN signaling (Figure 1 and Figure 2).

### 3.2. GSCs Expressed IFN Receptors and Reversibly Mediated Signaling after Chronic Exposure to Recombinant IFNs 

It has been reported that chronic exposure to IFNs reprogrammed the epigenome in melanoma cells via STAT1-dependent signaling [27]. Therefore, we tested whether chronic exposure to IFNs reprograms GSCs, constitutively activating their IFN signaling. First, we evaluated the expression of the IFN receptors, and GSCs expressed type I and type II IFN receptors at a transcriptional level at variable levels compared to the normal human brain (Appendix A). To confirm the functionality of the IFN receptors, we exposed the IFN-signaling-negative GSCs to IFN-β and IFN-γ for 48 h and evaluated IFN signaling. Both GSCs had dose-dependent increases in p-STAT1, t-STAT1, and MX1 expression (Figure 3A). Next, we evaluated intrinsic IFN reprogramming in IFN-negative GSCs after chronic exposure to low-dose IFN-γ (10 ng/mL) and IFN-β (50 ng/mL) for 2 weeks, followed by a washout period for 1 week. Chronic IFNs treatment showed significantly increased expression of p-STAT1, t-STAT1, and MX1 (Figure 3B–D). However, following a washout period of 1 week, the increased IFN signaling returned to basal level (Figure 3B–D). Unlike in melanoma cells [27], these results suggest a reversible activation of central nervous-system-specific IFN/STAT1 signaling.

### 3.3. GSCs Did Not Secrete IFNs, and Chronic IFN-γ Exposure Promoted Mesenchymal Signatures in GSCs 

To determine whether constitutive IFN signaling in GSCs was mediated by autocrine signaling mechanisms, we evaluated the secretion of IFN-γ, IFN-β, and inflammatory cytokines using ELISA in GSCs. We found that neither IFN-γ nor IFN-β was secreted in the CM of either low- or high-IFN-signaling GSCs (Appendix A). However, high-IFN-signaling GSCs secreted statistically significantly higher levels of IL-6 in 2 of 3 high-IFN-signaling GSCs compared to low-IFN-signaling GSCs (Appendix A). These results indicated that even GSCs with high IFN signaling secreted neither IFN-γ nor IFN-β. Therefore, we next evaluated whether IL-6 was responsible for differential activation of basal IFN signaling. We treated intrinsically IFN-negative GSCs with the equivalent doses of recombinant IL-6. We found that IL-6 treatment for 48 h did not increase the expression of p-STAT1, t-STAT1, and MX1 in GSC-7-11 and GSC-23 (Appendix A). This result indicated that, at least, IL-6 alone is not responsible for differential autocrine activation of IFN signaling in GSCs. 

IFN-γ is a strong inducer of IFN/STAT1 signaling, and is produced at high levels in the TME via T cells under immunosuppressive conditions [40]. We next tested whether chronic treatment with IFN-γ promoted the mesenchymal phenotype of GSCs. We treated GSC-7-11 and GSC-23 cells with IFN-γ (50 ng/mL) for 1 week, and the expression of a panel of the mesenchymal and stemness signature markers was evaluated by qPCR. Our results showed that IFN-γ treatment significantly increased the expression of *CD44*, *CD24*, *TIMP1*, *TIMP3*, and *STAT1* in GSC-7-11 (Figure 3E), as well as *CD44*, *TIMP1*, and *STAT1* in GSC-23 (Figure 3F). However, the effect of IFN-γ treatment on stemness genes (*CD133* and *SOX2*) showed variable expression in tested GSCs (Figure 3E,F). These results indicated that, in part, IFN-γ promoted the mesenchymal phenotype in GSCs with low IFN signaling. Thus, high basal IFN signaling in mesenchymal subtype GSCs may be reprogrammed by IFN-γ with or without other TME factors. 

### 3.4. Chronic Inhibition of IFN Signaling Reduces Cell Proliferation and Mesenchymal Signature in High-IFN-Signaling GSCs 

To determine whether high IFN signaling activated its downstream signaling pathways, we treated three GSCs with high IFN signaling with a JAK/STAT inhibitor ruxolitinib (500 and 1000 nM) for 72 h. Our results clearly showed that ruxolitinib remarkably reduced the expression of p-STAT1, t-STAT1, and MX1 in a dose-dependent manner in all three GSC lines (Figure 4A). These findings further confirmed that intrinsic high IFN signaling in GSCs is activated downstream from the IFN/STAT1 pathway. However, IL-6 (proinflammatory cytokine) treatment in low-IFN GSCs failed to activate the IFN/STAT1 signaling. 

To determine the contribution of IFN signaling to cell proliferation and mesenchymal properties of GSCs, we treated two GSC with low IFN signaling and three GSCs with high IFN signaling with a range of doses of ruxolitinib (25–5000 nM) for 3–5 days. Our results showed that 3–5 days of treatment with ruxolitinib did not reduce cell viability either in the GSCs with low IFN signaling or in the GSCs with high IFN signaling (Figure 4B,C and Appendix A). Next, we evaluated the influence of chronic ruxolitinib treatment (1 and 5 μM for 2 weeks) on the cell growth and proliferation in two high-IFN-signaling GSCs. Chronic ruxolitinib treatment significantly reduced cell growth and proliferation in a dose-dependent manner in both GSCs (Figure 4D,E). We next sought to further validate that ruxolitinib-mediated reduction in cell proliferation was associated with a decrease in the mesenchymal signature expression in these cells. We evaluated the expression of the mesenchymal signature using a panel of well-established biomarkers by qPCR in GSCs treated with ruxolitinib (5 µM) for 2 weeks. We found that ruxolitinib treatment significantly decreased the expression of *CD44*, *CD24*, *YKL40*, and *TIMP1* in GSC-17 (Figure 4F), as well as *CD44*, *CD24*, *YKL40*, *SERPINE1*, and *TIMP1* in GSC-20 (Figure 3G). Furthermore, ruxolitinib treatment decreased the expression of IFN signaling markers such as *IFI16*, *STAT1*, and *OAS2* in both GSCs (Figure 4F,G). However, chronic ruxolitinib treatment showed a differential effect on the expression of stemness genes (*SOX2* and *CD133*) in GSC-17 and GSC-20 (Figure 4F,G). These findings suggested that the IFN signaling contributed to the maintenance of mesenchymal phenotypes and proliferative properties of intrinsically high-IFN-signaling GSCs.

### 3.5. IFN-β Exposure Induced Apoptosis in GSCs with High Basal IFN/STAT1 without Modulating Stemness 

STAT1 is a key transcription factor involved in cell differentiation and apoptosis, and plays a central role in immune responses [14,15]. Thus, we reasoned that basal STAT1 protein levels might be a key determinant of apoptotic response to exogenous IFNs. We treated both STAT1-high and STAT1-low GSCs with IFN-β and IFN-γ for 72 h and evaluated the apoptotic response by c-PARP expression. Interestingly, we did not see the induction of the apoptosis signal (c-PARP expression) with either IFN-β or IFN-γ exposure in STAT1-low GSCs (Appendix A). To our surprise, we found that IFN-β induced apoptosis only in the STAT1-high GSCs (Figure 5A and Appendix A). However, IFN-γ did not induce apoptosis in STAT1-high GSCs (Appendix A). Moreover, IFN-β-mediated cell death was also evaluated by measuring cell viability with a range of doses (200–2000 IU/mL) in GSCs, and the results showed a decrease in cell viability in a dose-dependent manner in both GSCs (Figure 5B,C). In addition, we validated the IFN-β-mediated apoptotic cell death by annexin-V staining by flow cytometry. We found that IFN-β (1000 IU/mL) treatment for 48 h significantly increased early apoptotic cells (annexin-V positive) and late apoptotic cells (annexin-V and DAPI positive) in both GSCs (Figure 5D–F). These results suggested that basal STAT1 protein expression can be a determining factor for IFN-β-mediated apoptotic signaling in GSCs. 

To find out whether the IFN-β treatment modulated the stemness properties of GSCs, we evaluated the expression of stemness markers by qPCR in GSCs treated with IFN-β (1000 IU/mL) for 72 h. We did not see any notable effect of IFN-β treatment on stemness properties of either of the GSCs, as there was no significant change in the expression of *CD133*, *NANOG*, *SOX2*, and *CD44* (Appendix A). These results suggested that IFN-β induced apoptosis without modulating stemness properties.

### 3.6. Blockade of IFN/STAT1 with Ruxolitinib Reduced the IFN-β-Induced Apoptotic Cell Death

To determine whether IFN-β-induced apoptotic cell death was mediated via IFN/STAT1 signaling or other nonspecific signaling pathways, we blocked IFN/STAT1 signaling using ruxolitinib (1 µM) for 24 h in GSCs prior to IFN-β (1000 IU/mL) exposure, and apoptosis was evaluated. Our results showed that blockade of IFN/STAT1 with ruxolitinib significantly reduced IFN-β-induced apoptotic cell death in both the GSCs (Figure 5G). In addition, we confirmed these findings by evaluating the cell viability in the same experimental settings in both the GSCs, and results clearly showed that IFN-β-mediated reduction in cell viability was significantly rescued by pretreatment of ruxolitinib (Figure 5H,I). This finding confirmed that IFN-β-induced apoptotic cell death was mediated via IFN/STAT1 signaling. 

### 3.7. Single-Cell RNA-Seq Data Revealed Tumor-Cell IFN Signaling Was Associated with Mesenchymal Signatures 

To determine the possible contribution of *STAT1* in glioma progression, we analyzed *STAT1* expression in various datasets using the GlioVis platform [32]; *STAT1* expression was upregulated with increasing grades of glioma (Figure 6A–D). To determine the cell-type-specific activation of IFN signaling in tumor and nontumor cells, we analyzed publicly available human glioma single-cell RNA-seq (scRNA-seq) datasets from the three studies by Neftel et al. (GSE131928) [36], Darmanis et al. (GSE84465) [35], and Yu et al. (GSE117891) [37]. Our analyses showed that both tumor and nontumor cells expressed IFN signaling genes such as *STAT1*, *STAT2*, *ISG15*, *OAS1*, and *MX1* in all three datasets (Figure 6E,F and Appendix A). To find any association of IFN signaling in tumor cells with mesenchymal phenotypes, we chose five well-established signature genes of the mesenchymal phenotype: *CHI3L1*, *CD44*, *SERPINE1*, *TNC*, and *TIMP1*. We found that the mesenchymal signature genes were highly expressed in the same cluster where IFN signaling genes were upregulated in all three datasets (Figure 6E,F and Appendix A). In addition, our correlation analyses demonstrated a positive correlation between mesenchymal genes (*CHI3L1*, *CD44*, and *SERPINE1*) and IFN signaling genes (Figure 6G–I and Appendix A). These findings indicated that tumor-cell IFN signaling was associated with mesenchymal phenotypes of GBM. Finally, we evaluated the association of *STAT1* expression with survival outcomes of glioma/GBM patients. We found that high *STAT1* expression was negatively associated with overall survival (Appendix A). Moreover, Chinese Glioma Genome Atlas (CGGA) data showed that high *STAT1* expression in GBM mesenchymal tumors was associated with poor overall survival (Figure 6J). However, there was no association between patients’ survival and *STAT1* expression in other subtypes of GBM tumors in the TCGA and CGGA datasets (Appendix A).

### 3.8. STAT1 Knockdown Reduced the IFN-β-Induced Apoptotic Cell Death

Since ruxolitinib is a pan-JAK-STAT pathway blocker, its pharmacological effect might be mediated by inhibiting other STAT proteins [41]. To further determine IFN-β-induced cell death mediated by STAT1 signaling in GSCs, we knocked down (KD) the STAT1 protein in two high STAT1 expressing GSCs (GSC17 and GSC20) by CRISPR/Cas9 technology. In both GSCs, the STAT1 KD was confirmed by Western blotting (Figure 7A) and by qPCR (Appendix A). CRISPR/Cas9-mediated STAT1 KD did not alter the expression of other STATs such as *STAT2* and *STAT3* (Appendix A). To validate reduced IFN-β-induced apoptosis and IFN/STAT1 signaling by ruxolitinib treatment, we treated vector control and STAT1 KD GSC17 and GSC20 with IFN-β (1000 IU/mL) for 48 h. Our results showed that STAT1 KD significantly reduced the IFN-β-mediated cell death (assessed by c-PARP levels) and the expression of p-STAT1 and t-STAT1 in both GSCs (Figure 7B). STAT1 KD rescued the IFN-β-induced reduction in cell viability in both cell lines (Figure 7C,D). Next, we evaluated the apoptotic cell death in the STAT1 KD cells using annexin-V and DAPI staining by flow cytometry. Results showed that STAT1 KD significantly reduced the IFN-β-induced apoptotic cell death in both cell lines (Figure 7G–I). Furthermore, we confirmed the specificity of STAT1-mediated signaling by evaluating the expression of *STAT1*, *STAT2*, and *OAS2* in the same experimental settings in both GSCs, and results showed that STAT1 KD significantly reduced IFN-β-mediated expression of *STAT1* without altering the expression of *STAT2* and *OAS2* as compared to vector control (Figure 7E,F). These experiments confirmed that IFN-β-induced apoptotic cell death was mediated via STAT1 signaling in GSCs with a high intrinsic IFN signaling.

## 4. Discussion

IFN signaling is known to play a critical role in immunologic surveillance, immune response, and multigenic resistance to immunotherapies in many cancers, including GBM [27,40,42]. However, the molecular mechanisms of IFN-mediated effects on glioma cells and on distinct GSC populations have remained unclear. Our findings demonstrated that distinct GSCs and GBM tumors exhibited differential IFN signaling, which is tightly correlated with mesenchymal phenotypes, and that basal IFN/STAT1 was a critical factor in IFN-β-mediated GSC cell death. A limitation of TCGA analyses is the inability to determine whether RNA species are from tumor cells or from infiltrating host cells. However, a similar analysis in a cohort of GSCs resembling the TCGA-inferred molecular subtypes confirmed the intrinsic IFN signature in GSCs. The upregulated expression of p-STAT1, t-STAT1, and MX1 under basal conditions in a subset of GSCs further confirmed the in-silico analyses. 

Our results agreed with a study by Doucette et al., which highlighted the enrichment of proinflammatory cytokines and IFN genes within the mesenchymal subtype of GBM [43]. In that study, the mesenchymal signature was associated with poor prognosis and survival; thus, we cannot rule out that IFN secretion might stem from the resident immune cells, such as microglia or partially differentiated non-stem-like cells [43]. The high levels of IFN signature in mesenchymal GSCs might induce IFN-stimulated gene transcription within the TME. In our study, acute exposure of low-IFN-signaling GSCs to IFN-β and IFN-γ increased the expression of p-STAT1, t-STAT1, and MX1 compared with untreated cells (Figure 3). Furthermore, activation of IFN signaling via acute exposure to IFNs (ligands) also confirmed that IFN signaling was mediated via their specific receptors. It has also been reported that autocrine IFN signaling is constitutively active in glioma tumors and glioma non-stem cells [18], and malignant glioma cells expressed high STAT1, which influences tumor cell proliferation, migration, and invasion [44]. Furthermore, our survival analyses revealed that high type I and type II IFN signaling were associated with poor survival outcomes, suggesting a negative impact of constitutively active IFN signaling in these tumors. 

Both GSCs and GBM tumors showed high IFN signaling in mesenchymal subtypes, which might be due to continuous exposure to cytokines and IFNs secreted by tumor cells and/or infiltrating nontumor cells or immune cells within the TME. Chronic exposure to IFNs has been observed to reprogram melanoma cells via a STAT1-dependent mechanism [27]. To validate the similar phenomena in GBM, we chronically activated type I and type II IFN signaling in GSCs with no to low basal levels of IFN signaling using recombinant IFNs for 2 weeks followed by 1 week of washing (no treatment). Our data showed that both type I and type II IFN signaling activation was unable to reprogram GSCs, as revealed by the reversal of the expression of IFN-stimulated genes to normal levels after a 1-week washout of IFN treatment. 

We observed the interruption of endogenous active IFN signaling in three mesenchymal GSC cell lines treated with ruxolitinib remarkably reduced the expression of IFN-signaling proteins in a dose-dependent manner. This indicated that constitutive IFN signaling in mesenchymal GSCs and GBM tumors operates primarily through IFN receptors, rather than through other cytokine signaling or receptors such as IL-6. Instead, Wang et al. have reported that IL-6 signaling contributed to GSC survival and tumor growth [45], and that perturbation of IL-6 signaling in GSCs attenuated STAT3 activation, which is a downstream mediator of IL-6-mediated pro-survival signaling in GSCs. Although we found that a subset of mesenchymal GSCs (GSC-20 and GSC-28) secreted IL-6, it did not induce IFN/STAT1 signaling in GSCs, because IL-6 primarily mediated STAT3 signaling. We found that chronic ruxolitinib treatment significantly reduced cell proliferation and mesenchymal signatures in a dose-dependent manner in the GSCs with high intrinsic IFN signaling. Moreover, we did not observe any significant changes on the stemness genes expression with IFN-β treatment in GSCs. This observation was an agreement with similar findings reported by Du et al. that there was no major effect on the stemness genes with type I IFN (IFN-α) exposure in GBM stem cell lines [46]. Further, Du et al. reported that IFN-α induced transient STAT3 activation in GSCs and reduced the cell proliferation without modulating the stemness markers. Nevertheless, we observed an inconsistent modulation of the stemness genes (*CD133* and *SOX2*) in GSCs with the chronic IFN-γ or ruxolitinib treatment. These differential effects on stemness genes after the chronic IFN-γ or ruxolitinib treatment might have attributed to the GSCs plasticity and heterogeneity [47], which are modulated by several signaling cascades, including JAK-STATs (STAT1 and STAT3) [22,48]. In addition, the activation of STAT proteins is regulated by cytokine and growth factor receptors and by negative-feedback signaling mechanisms [49].

Our scRNA-seq analysis demonstrated that both tumor and nontumor cells expressed the IFN signaling genes. Furthermore, all three studies’ datasets showed that high expression of mesenchymal signature genes was in the same cluster in which IFN signaling genes were upregulated. These observations suggested that the TME can induce IFN signaling in tumor cells, which contributes to the malignant mesenchymal GBM phenotype. Previously, we have shown that STAT1/IRF-1 signaling was involved in the development of bevacizumab resistance (often mesenchymal), and that genetic inhibition of IRF1 increased apoptosis in bevacizumab-treated glioma cells [50]. Recently, Giangos et al. reported that GSCs establish a myeloid mimicry via an epigenetic program to drive a myeloid-enriched TME, thereby enabling immune evasion and tumor progression [51]. They further suggested that prolonged exposure to IFN-γ facilitated the acquisition of a protumorigenic TME, which can drive transcriptional changes similar to a mesenchymal phenotype [51]. Our findings suggest that IFN-γ exposure to low-IFN GSCs promoted a mesenchymal phenotype in GSCs, which might contribute to an aggressive phenotype, as is observed in recurrent GBM. Activation of IFN-related pathways may lead to aggressive tumor progression; however, the specific mechanisms related to the development of a mesenchymal phenotype need to be further investigated. 

Finally, we tested whether differential IFN/STAT1 signaling had any role in cell death and treatment resistance in GBM. Exposure of the low- and high-IFN-signaling GSCs to recombinant IFN-β specifically induced apoptosis only in intrinsically high-IFN/STAT1 GSCs, while IFN-γ failed to induce death signaling in either low- or high-IFN/STAT1 GSCs. Thus, the role of IFN-β (type I) in regulating GSC properties and apoptotic signaling may depend on molecular subtype or inherent basal levels of STAT1 protein. We further confirmed that IFN-β exposure induced cell death via STAT1-mediated signaling by employing the pharmacological inhibitor of the JAK/STAT pathway (ruxolitinib), as well as STAT1 KD. Our results were in line with a previous study showing that IFN-β mediated proapoptotic signaling in glioma cells to exogenous death ligands, which indicated distinct pathways of cell death [52]. Zhu et al. validated a prognostic model for the interferon signature and treatment response in GBM and lower grade glioma patient’s data from six cohorts and showed an in increased IFN signaling in GBM. Furthermore, Zhu et al. developed a five-gene-based IFN signature that could serve as independent indicator for unfavorable prognosis [53]. Interestingly, an overexpression of IFN-β is associated with the survival benefit in the low-risk group as compared to high-risk-group patients. Another report demonstrated that IFN-β gene therapy mediated by an intracranial adenoviral vector significantly reduced the tumor burden and doubled the median survival in highly migratory GBM in a syngeneic mouse model [54]. In addition, IFN-β has shown promise as a therapeutic agent in combination with temozolomide in a trial for newly diagnosed primary GBM [55]. Another pilot clinical trial in recurrent glioma showed antitumor activity for IFN-β and was well tolerated up to 40 weeks [56]. In contrast, Sgorbissa et al. reported the differential regulation of type I IFN signaling in GBM cells and resistance to apoptosis in response to IFN-α treatment [57]. 

STAT1 is a well-known transcription factor that plays a critical role in tumor development, cell growth, proliferation, and apoptotic cell death [15]. In addition, STAT1 expression in malignant glioma cells regulates tumor cell proliferation, migration, and invasion, and STAT1 level in human GBM tissues has been proposed as a novel prognostic biomarker [44]. Our results suggest constitutively high IFN signaling in mesenchymal GSCs and GBM tumors. Therefore, basal STAT1 expression might contribute to the IFN-β-induced apoptosis in GSCs with high IFN signaling. However, additional studies are warranted to describe the molecular mechanisms of how the basal STAT1 protein levels modulate apoptosis in GBM. IFN signaling is involved in direct tumor-cell killing and/or indirect antitumor immune responses of targeted immunotherapies and oncolytic viruses in many cancers, including brain tumors [10,58]. Furthermore, chronic activation of IFN signaling in tumor cells facilitated resistance to immune checkpoint blockade via multiple inhibitory pathways [27]. Despite the central and indirect roles of IFN signaling in antitumor response induced by various therapies, the function of IFN signaling in brain tumors has largely remained unexplored. Here, we provided insights into the IFN signaling in GSCs that may improve clinical trial design and the development of oncolytic viruses and other immunotherapies for GBM patients.

## 5. Conclusions

In summary, our findings demonstrated that GSCs and GBM tumors exhibited differential cell-intrinsic type I and type II IFN/STAT1 signaling, and that high IFN signaling was associated with mesenchymal phenotype and poor survival outcomes. Chronic exposure to recombinant IFN proteins reversibly activated type I and type II signaling in GSCs. The IFN-β exposure specifically induced apoptosis in GSCs with intrinsically high IFN/STAT1 signaling. Furthermore, genetic inhibition or pharmacological inhibition of STAT1 in GSCs reduced the IFN-β-induced cell death and IFN/STAT1 signaling. Therefore, the basal STAT1 levels might be substantially contributing to IFN-β-mediated cell death in GSCs. Our study provides evidence for the possibility of targeting IFN signaling in a specific group of GBM patients. Indeed, the clinical use of IFN-β has been evaluated in many clinical studies [59,60], and its safety and tolerance have already been demonstrated, making it a promising candidate for adjuvant GBM therapy. Basal STAT1 levels may be a novel predictive/prognostic biomarker for screening patients that may benefit from IFN-β treatment.

## Figures and Tables

**Figure 1 cancers-13-05284-f001:**
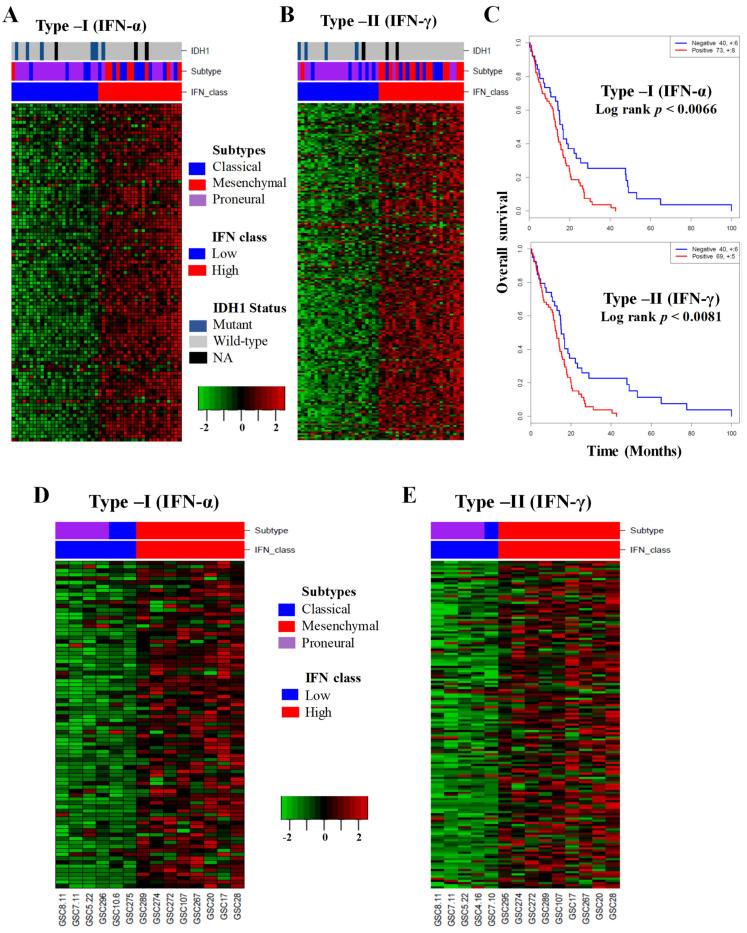
Analysis of type I and type II IFN signaling gene expression of GBM patients’ tumors in the TCGA database and GSCs. (**A**,**B**) Heat maps of TCGA RNA-seq data sets of GBM tumors. The normalized gene level for RNA-seq data was analyzed separately for the hallmark genes representing type I and type II IFN-low and -high signaling in tumors. (**C**) Survival analysis of IFN-low and -high signaling of GBM patients. (**D**,**E**) Heat maps of the type I and type II IFN signaling genes in the RNA-seq data in a cohort of GSCs.

**Figure 2 cancers-13-05284-f002:**
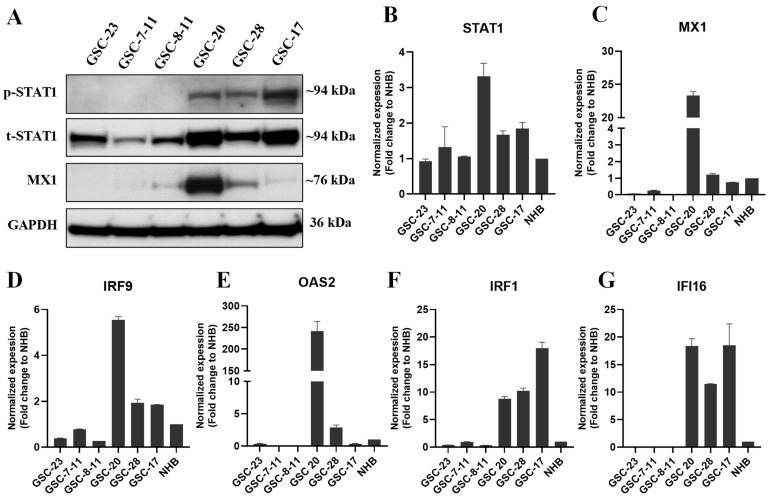
Basal protein and mRNA expression of IFN signaling in a cohort of GSCs. (**A**) WB analysis of p-STAT1, t-STAT1, and MX1 in whole-cell lysates (WCLs) of a subset of low- and high-IFN-signaling GSCs. Original blots see Appendix A. (**B**–**G**) The mRNA expression of *STAT1*, *MX1*, *IRF9*, *OAS2*, *IRF1*, and *IFI16* in GSCs normalized to normal human brain (NHB).

**Figure 3 cancers-13-05284-f003:**
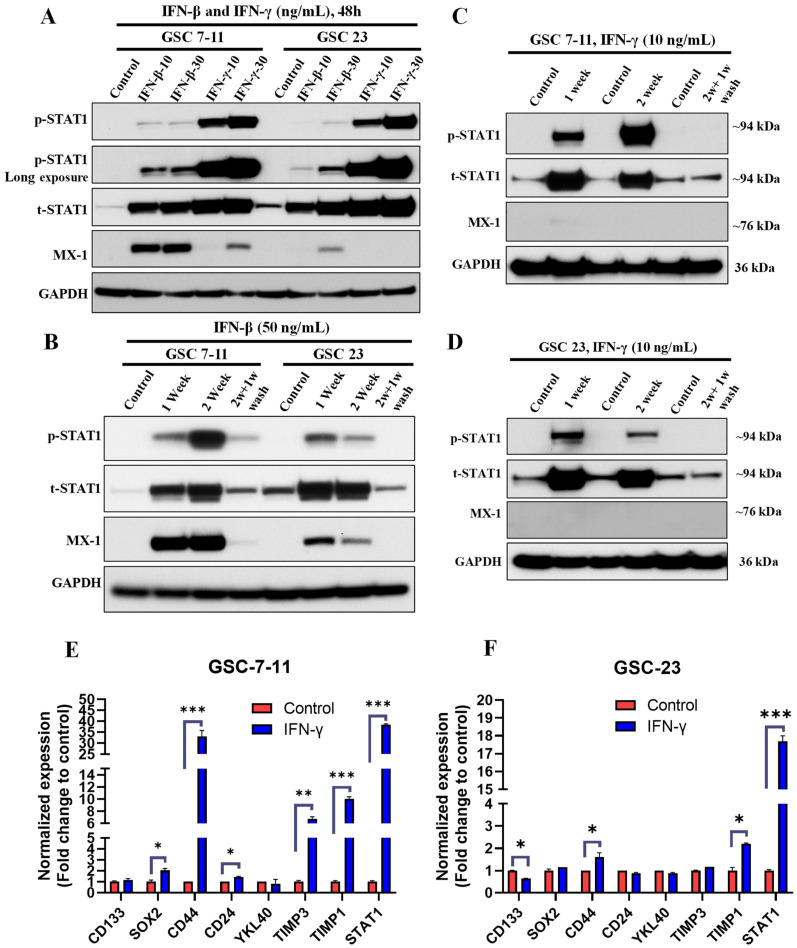
Chronic IFN exposure reversibly activated IFN signaling, and IFN-γ exposure promoted mesenchymal signatures in GSCs. (**A**) Representative WB of IFN/STAT1 signaling in WCLs of GSC-7-11 and GSC-23 treated with recombinant IFN-β and IFN-γ (10 and 30 ng/mL) for 48 h. (**B**–**D**) WB of IFN/STAT1 signaling in WCLs of GSC-7-11 and GSC-23 treated with recombinant IFN-β (50 ng/mL) and IFN-γ (10 ng/mL) for 2 weeks, followed by a washout period for 1-week. The IFN-treatment-containing media was replaced every 72 h. Original blots see Appendix A. (**E**,**F**) mRNA expression of mesenchymal signature genes in GSCs treated with IFN-γ (50 ng/mL) for 1 week. *** *p* < 0.001, ** *p* < 0.01, and * *p* < 0.05 compared to untreated cells.

**Figure 4 cancers-13-05284-f004:**
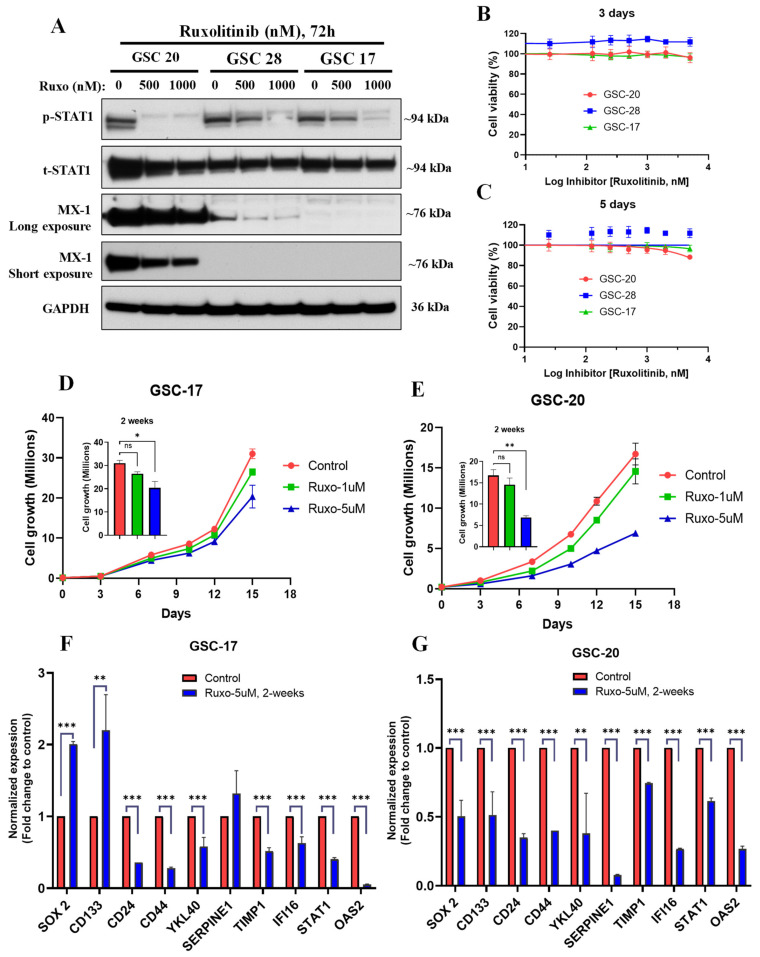
Chronic inhibition of IFN signaling reduced cell proliferation and mesenchymal signature. (**A**) Representative WB of IFN/STAT1 signaling proteins in WCLs of GSCs treated with ruxolitinib (500 and 1000 nM) for 72 h. Original blots see Appendix A. (**B**,**C**) Cell viability and proliferation of GSCs treated with ruxolitinib (25–5000 nM) for 3 days and 5 days. (**D**,**E**). Cell proliferation of GSCs treated with ruxolitinib (1 and 5 µM) for 2 weeks. Ruxolitinib-treatment-containing media was changed every 72 h. (**F**,**G**) mRNA expression of mesenchymal and IFN genes in GSCs treated with ruxolitinib (5 µM) for 2 weeks. *** *p* < 0.001, ** *p* < 0.01, and * *p* < 0.05 compared to untreated cells.

**Figure 5 cancers-13-05284-f005:**
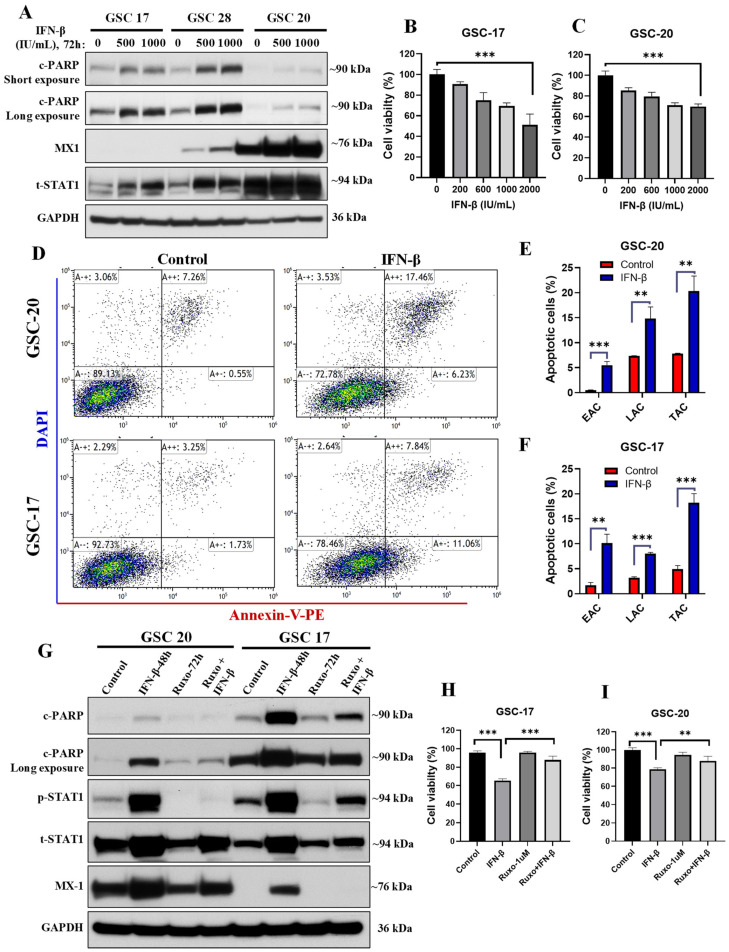
IFN-β exposure induced apoptosis in GSCs with high basal IFN/STAT1. (**A**) Representative WB of c-PARP and IFN/STAT1 signaling WCLs of GSCs treated with IFN-β (500–1000 IU/mL) for 72 h. Original blots see Appendix A. (**B**,**C**) Cell viability of GSCs treated with IFN-β (200–2000 IU/mL) for 72 h. (**D**–**F**) Apoptosis analysis by flow cytometry in GSCs treated with IFN-β (1000 IU/mL) for 48 h. EAC: early apoptotic cells, LAC: late apoptotic cells, TAC: total apoptotic cells. (**G**) Representative WB of apoptosis and IFN/STAT1 signaling in WCLs of GSCs treated with ruxolitinib (1 μM) for 24 h before IFN-β (1000 IU/mL) exposure for 48 h. Original blots see Appendix A. (**H**,**I**) Cell viability of GSCs treated as mentioned in (**G**). *** *p* < 0.001 and ** *p* < 0.01 compared to untreated cells.

**Figure 6 cancers-13-05284-f006:**
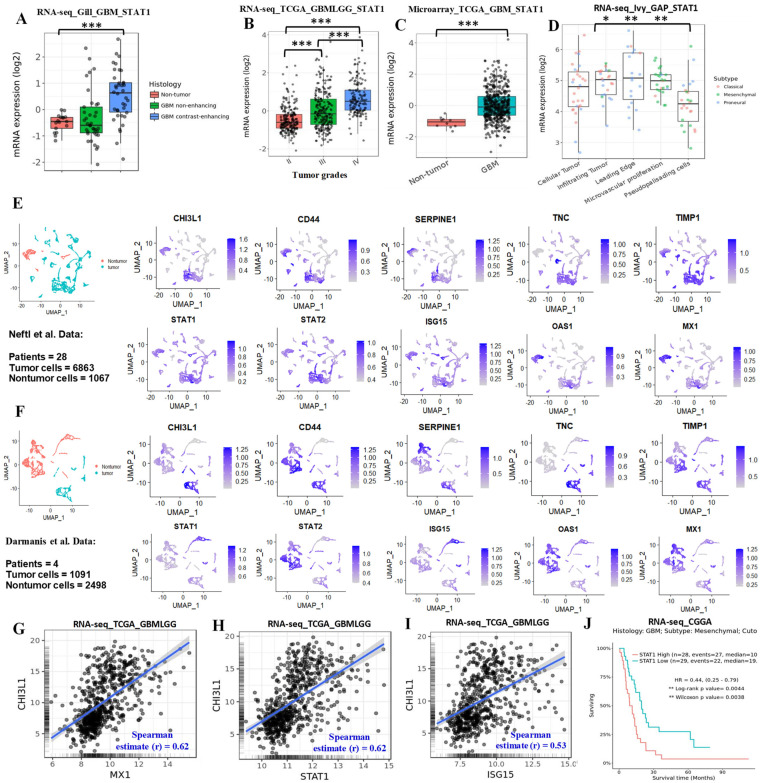
IFN/STAT1 expression associated with glioma progression and mesenchymal signature. (**A**–**D**) Expression of STAT1 in glioma tumor in the various datasets analyzed using the GlioVis platform. *** *p* < 0.001, ** *p* < 0.01, and * *p* < 0.05. (**E**,**F**) The scRNA-seq analysis of the selected genes for the IFN and mesenchymal signatures in tumor and nontumor cells; data were obtained from Neftel et al. (GSE131928) and Darmanis et al. (GSE84465), respectively. (**G**–**I**) Correlation analyses of CHI3L1 with MX1, STAT1, and ISG15 expression in glioma tumors (TCGA data) analyzed using the GlioVis platform. (**J**) Survival analysis of the low and high expression of STAT1 in the mesenchymal GBM tumor specimens in the CGGA dataset analyzed using the GlioVis platform.

**Figure 7 cancers-13-05284-f007:**
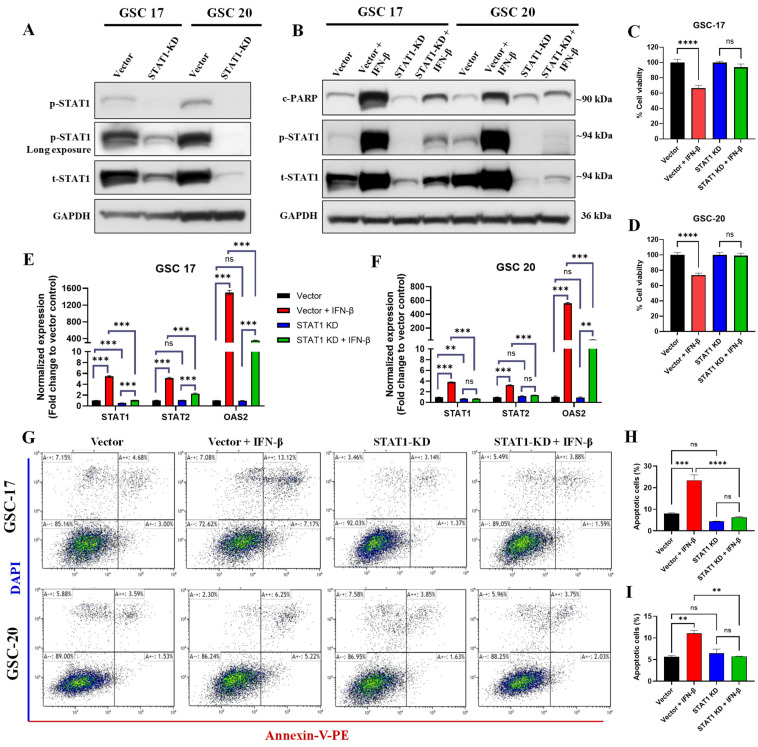
CRISPR/Cas9-mediated STAT1 KD reduced the IFN-β-induced cell death in GSCs. (**A**) Representative WB of the basal expression of p-STAT1 (Y701) and t-STAT1 in WCLs of vector control and STAT1 KD GSC17 and GSC20. Original blots see Appendix A. (**B**) Representative WB of c-PARP, p-STAT1, and t-STAT1 in WCLs of vector control and STAT1 KD GSC17 and GSC20 treated with IFN-β (1000 IU/mL) for 48 h. (**C**,**D**) Cell viability of vector control and STAT1 KD GSC17 and GSC20 treated with IFN-β (1000 IU/mL) for 48 h. (**E**,**F**) The expression of *STAT1*, *STAT2,* and *OAS2* in vector control and STAT1 KD GSC17 and GSC20 treated with IFN-β (1000 IU/mL) for 48 h. (**G**) Apoptosis analysis (annexin-V and DAPI staining) by flow cytometry in vector control and STAT1 KD treated with IFN-β (1000 IU/mL) for 48 h. (**H**,**I**) Bar graph of total apoptotic cells in vector control and STAT1 KD GSC17 and GSC20 treated as mentioned in (**G**), respectively. **** *p* < 0.0001, *** *p* < 0.001, ** *p* < 0.01; ns, not significant (*p* > 0.05).

## Data Availability

Publicly available datasets were analyzed in this study. The data can be found here: https://gdac.broadinstitute.org/, accessed on 22 February 2021; http://cbioportal.org, accessed on 22 February 2021. The single-cell RNA-seq data were analyzed from Darmanis et al. (GSE84465) [35], Neftel et al. (GSE131928) [36], and Yu et al. (GSE117891) [37] studies.

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
