# Peer review of "Intrinsic Interferon Signaling Regulates the Cell Death and Mesenchymal Phenotype of Glioblastoma Stem Cells"

_cancers, 2021, doi:10.3390/cancers13215284_

Round 1
Reviewer 1 Report
The authors have addressed the concerns raised.
Reviewer 2 Report
Sabbir Khan and co-workers investigate the role of IFN signaling in GBM stem-like cells, thus correlating IFN/STAT1 signaling with GBM cell proliferation and GBM mesenchymal signatures.
The authors addressed the reviewer concerns by using a CRISPR/Cas9 based STAT1 knockdown in GSCs. The mechanism is well defined and described and the specificity of the role of STAT1 in the response to INF-β administration was experimentally confirmed, thus strengthening the use of ruxolitinib and or INF-β.
This manuscript is a resubmission of an earlier submission. The following is a list of the peer review reports and author responses from that submission.
Round 1
Reviewer 1 Report
The work by Khan and cols., reports the effect of IFN-gamma on the mesenchymal phenotype/cell proliferation (poor patient survival), and IFN-beta-triggered cell death in GBM-GSCs. They show that STAT1-dependent signaling seems to play a role in either IFNgamma- or IFNbeta-associated cellular effects. The authors employ ruxolitinib, a JAK1/JAK2 inhibitor, to state that IFN/STAT1 axis governs both mesenchymal transformation/cell proliferation and cell death. This is a well performed study. The data presented are in good quality and, in general, supportive to the main conclusion. I do have some comments/suggestions:
- The authors are focused in STAT1 signaling. Is there any reason why the authors do not study/discuss STAT2 or STAT3? These two are also assumed to be activated by IFN. In this sense, for example, Du and colleagues (Biochem Biophys Res Commun. 2017 Sep 16;491(2):343-348.) reported that IFN induces a transient STAT3-mediated signal in GBM/GSCs. The authors should contrast and discuss the results obtained with the data shown by Du et al.
- The discussion would have to be updated (f. ex., Oncoimmunology. 2019; 8(9): e1621677).
- To corroborate the main conclusion, the authors could silence (or knockout) STAT1 in either high or low STAT1-expressing GSCs, and perform key experiments to specifically demonstrate the need of IFN/STAT1.
- Figure 5, graphs E and F, spell EAC, LAC and TAC.
- What is the point of abbreviating “immunotherapies” by “IMTs”?
Author Response
We would like to thank both the reviewers for their positive notes and the constructive comments, which helped us to improve the manuscript. Below is the point-by-point reply to the specific comments.
Reviewer #1
The work by Khan and cols., reports the effect of IFN-gamma on the mesenchymal phenotype/cell proliferation (poor patient survival), and IFN-beta-triggered cell death in GBM-GSCs. They show that STAT1-dependent signaling seems to play a role in either IFNgamma- or IFNbeta-associated cellular effects. The authors employ ruxolitinib, a JAK1/JAK2 inhibitor, to state that IFN/STAT1 axis governs both mesenchymal transformation/cell proliferation and cell death. This is a well performed study. The data presented are in good quality and, in general, supportive to the main conclusion. I do have some comments/suggestions:
Comment-1: The authors are focused in STAT1 signaling. Is there any reason why the authors do not study/discuss STAT2 or STAT3? These two are also assumed to be activated by IFN. In this sense, for example, Du and colleagues (Biochem Biophys Res Commun. 2017 Sep 16;491(2):343-348.) reported that IFN induces a transient STAT3-mediated signal in GBM/GSCs. The authors should contrast and discuss the results obtained with the data shown by Du et al.
Reply: First we would like to thank the reviewer for his positive notes and the comments and suggestions. We agree with the reviewer that the current study didn’t focus much attention on STAT2/STAT3 signaling. The current study aimed to investigate the role of cancer-cell intrinsic interferon (IFN) signaling in GSCs and GBM tumors and particularly focused on the less studied role of STAT1, a central mediator molecule for the IFNs signaling . As the reviewer pointed out we included the study by Du et al and the role of STAT3 in the revised manuscript (highlighted text in the lines 470-482, and references 22 and 45-48).
Comment-2: The discussion would have to be updated (f. ex., Oncoimmunology. 2019; 8(9): e1621677).
Reply: Thanks for pointing out the missing reference. We have updated the discussion and referenced the Zhu et al study (Oncoimmunology. 2019; 8(9): e1621677) in the revised manuscript. [highlighted text in the lines 508-513 and updated reference #52]
Comment-3: To corroborate the main conclusion, the authors could silence (or knockout) STAT1 in either high or low STAT1-expressing GSCs,and perform key experiments to specifically demonstrate the need of IFN/STAT1.
Reply: We would like to thank the reviewer for this suggestion. We evaluated STAT1 and its inhibition by Ruxolitinib with Western blot and real-time RT-PCR in all the experiments, which clearly demonstrated the involvement of STAT1 in cell proliferation, cell death and promoting mesenchymal phenotype. However, we agree with the reviewer concern for specificity of STAT1-mediated signaling in the current study and the potential off-target effects of ruxolitinib on other STAT proteins, like STAT2 and STAT3. We have discussed the possible contribution of other STAT isoforms in the revised manuscript with additional references. In addition, a statement has been added in the summary/conclusion of the revised manuscript regarding the need of future studies to delineate the contribution of STAT1 as well as other STAT proteins in the revised manuscript (highlighted text in the lines 470-482 and 550-553). Based on your suggestions, we hope to address the mechanistic role of STAT1 signaling in glioma diseases and immunotherapy resistance with future studies.
Comment-4: Figure 5, graphs E and F, spell EAC, LAC and TAC.
Reply: Thank you for your comment. The legend of the figure 5 has been updated as per the suggestion. [highlighted text in the line 369]
Comment-5: What is the point of abbreviating “immunotherapies” by “IMTs”?
Reply: We updated the revised manuscript without abbreviation. [highlighted text in the revised manuscript]
Reviewer 2 Report
In this study, Sabbir Khan and co-workers investigate the role of the intrinsic IFN signaling in glioblastoma tumours and its correlation with Glioblastoma specific subtype and patients’ outcome.
The study is well designed, and the experiment are properly conducted each with appropriate controls, however there are some points to be further clarify:
1 – Relation between IFN intrinsic signalling and GBM stemness. The authors here investigate the role of IFN-β in modulating the expression levels of stemness genes in GBM cells (Suppl fig S5) and they do not find any differences between control and treated cells. Conversely, figure 3E-F and figure 4 F-G clearly show a significant differential regulation in these stemness genes that the authors do not even mention in the manuscript. Could you please verify by other techniques (i.e. flow cytometry or WB) the specific expression of CD133, Sox2, Nanog etc after IFN pathway modulation? There could be any correlation also with proliferation and/or self-renewal?
2- Ruxolitinb related data are thus interesting from a therapeutic point of view,. There is any synergistic action between Ruxolitinib and Temozolomide?
3- Authors here demonstrated that Ruxolitinib administration impacts on cell proliferation and mesenchymal genes expression. However, Ruxolitinb is quite a “dirty drug” since its first targets are JNK proteins that could also modulate cancer cells phenotype. It will be interesting investigate if the mechanism is STAT1 specific by a STAT1 specific silencing.
Minor consideration:
Check for typing errors.
Author Response
We would like to thank both the reviewers for their positive notes and the constructive comments, which helped us to improve the manuscript. Below is the point-by-point reply to the specific comments.
Reviewer #2
In this study, Sabbir Khan and co-workers investigate the role of the intrinsic IFN signaling in glioblastoma tumours and its correlation with Glioblastoma specific subtype and patients’ outcome.
The study is well designed, and the experiment are properly conducted each with appropriate controls, however there are some points to be further clarify:
Comment-1: – Relation between IFN intrinsic signalling and GBM stemness. The authors here investigate the role of IFN-β in modulating the expression levels of stemness genes in GBM cells (Suppl fig S5) and they do not find any differences between control and treated cells. Conversely, figure 3E-F and figure 4F-G clearly show a significant differential regulation in these stemness genes that the authors do not even mention in the manuscript. Could you please verify by other techniques (i.e. flow cytometry or WB) the specific expression of CD133, Sox2, Nanog etc. after IFN pathway modulation? There could be any correlation also with proliferation and/or self-renewal?
Reply: First we thank the reviewer for their appreciation of the current study. Regarding the comment about the IFN signaling and stemness, as the reviewer pointed out we did not find any effect on stemness gene expression with IFN-β treatment in GSCs. This is consistent with the similar findings reported by Du et al.,( Biochemical and Biophysical Research Communications, (2017) 491:343-348) where they did not see any major effect on stemness genes with type I IFN (IFN-α) exposure in two GBM stem cell lines. However, in the current study we observed variable expression of stemness gene (i.e CD133 and SOX2) expression in GSCs with the chronic treatment of IFN-γ or ruxolitinib. We believe this might be because stemness marker expression in GSC or GBM tumors is plastic and heterogenous in nature and regulated by multiple signaling pathways and/or contribution of other STAT proteins (Dirkse et al.,Nature Communications, (2019) 10:1787). Therefore, we believe this differential modulation of the stemness genes (CD133 and SOX2) in GSCs with the IFN-γ and ruxolitinib exposure might be due to activation of feedback mechanisms under chronic stimulation or inhibition of IFN signaling. As the reviewer suggested we have updated the results section and discussed this differential modulation of the stemness genes and possible contribution of feedback mechanisms as well as contribution of additional STATs proteins in the revised manuscript (highlighted text in the line lines 296-297, 327-329, and 470-482).
Comment-2:- Ruxolitinb related data are thus interesting from a therapeutic point of view,. There is any synergistic action between Ruxolitinib and Temozolomide?
Reply: We thank the reviewer for this interesting thought. We have not tested this combination in the current study.
Comment-3:- Authors here demonstrated that Ruxolitinib administration impacts on cell proliferation and mesenchymal genes expression. However, Ruxolitinb is quite a “dirty drug” since its first targets are JNK proteins that could also modulate cancer cells phenotype. It will be interesting investigate if the mechanism is STAT1 specific by a STAT1 specific silencing.
Reply: We thank the reviewer for raising the off-target effects of ruxolitinib. We agree with the reviewer concern for the pan-JAK-STATs pathway blocked by ruxolitinib and its specificity on STAT1-mediated signaling. Although, ruxolitinib is shown to block its primary targets JAK1 and JAK2 and blocked the downstream cascades of JAK-STATs signaling node, possible off-target effects on of other STATs protein such as STAT2 and STAT3 cannot be ruled out in the current findings. Briefly, we have discussed the possible contribution of other STATs isoform in the revised manuscript with additional references (highlighted text in the lines 470-482 and 550-553). Based on your suggestions, we hope to address the mechanistic role of STAT1 signaling in glioma diseases and immunotherapy resistance with future studies.
Minor consideration:
Comment-4: Check for typing errors.
Reply: Thank you for suggestion. We have carefully reviewed and corrected the entire manuscript for typos mistakes by our institutes scientific editing team.
Round 2
Reviewer 1 Report
The use of compounds such as Ruxolitinib does not sustain the specific role of STAT1 claimed by the authors. The authors have not performed additional experiments to corroborate their hypothesis and conclusions.
Reviewer 2 Report
In this study, Sabbir Khan and co-workers investigate the role of the intrinsic IFN signaling in glioblastoma cells thus correlating IFN action to the intracellular role of STAT1.
The authors here did not full fill the experimental request of the reviewer, however they implemented the manuscript discussing the concerns the reviewer raised about they conclusion.